# Multimodal sensory integration in single cerebellar granule cells in vivo

Taro Ishikawa[1,2†], Misa Shimuta[1,2†], Michael Häusser[1*]

[1]Wolfson Institute for Biomedical Research, Department of Neuroscience, Physiology and Pharmacology, University College London, London, United Kingdom; [2]Department of Pharmacology, Jikei University School of Medicine, Tokyo, Japan

**Abstract** The mammalian cerebellum is a highly multimodal structure, receiving inputs from multiple sensory modalities and integrating them during complex sensorimotor coordination tasks. Previously, using cell-type-specific anatomical projection mapping, it was shown that multimodal pathways converge onto individual cerebellar granule cells (Huang et al., 2013). Here we directly measure synaptic currents using in vivo patch-clamp recordings and confirm that a subset of single granule cells receive convergent functional multimodal (somatosensory, auditory, and visual) inputs via separate mossy fibers. Furthermore, we show that the integration of multimodal signals by granule cells can enhance action potential output. These recordings directly demonstrate functional convergence of multimodal signals onto single granule cells.

*For correspondence:
m.hausser@ucl.ac.uk

†These authors contributed equally to this work

## Introduction

Integrating multimodal sensory signals is one of the fundamental operations performed by the brain. The midbrain and cerebral association cortex (*Stein and Stanford, 2008*), and the cerebellum receive and process sensory signals of various modalities (*Snider and Stowell, 1944*; *Azizi and Woodward, 1990*; *Gao et al., 1996*; *Sobel et al., 1998*). Although it is well known that each granule cell in the mammalian cerebellum receives excitatory synaptic inputs from on average four mossy fibers (*Eccles et al., 1967*; *Jakab and Hamori, 1988*), it is important to determine whether multisensory integration takes place already on the level of the granule cells, which form the input layer, or only in the downstream neurons, including Purkinje cells. A recent study using projection mapping has shown that some granule cells receive mossy fibers from two areas in the brainstem (the basilar pontine nucleus and the external cuneate nucleus), providing morphological evidence of multimodal convergence in single granule cells (*Huang et al., 2013*; see also *Chabrol et al., 2015*). However, in vivo single cerebellar granule cells in cats have been shown to only respond to stimulation of a single modality (*Jörntell and Ekerot, 2006*; *Spanne and Jörntell, 2015*). Therefore, to determine whether functional multimodal convergence is common we made whole-cell patch-clamp recordings in vivo from single granule cells and directly tested responsiveness to sensory stimulation of different modalities.

## Results

### Granule cell responses to different sensory modalities

We made patch-clamp recordings from single cerebellar granule cells of rats in vivo to examine their responses to auditory, visual and somatosensory stimulation. We selected crus I and II and the dorsal paraflocculus of the cerebellum because previous studies suggested that these areas might receive

multisensory inputs (*Azizi and Woodward, 1990*; *Huang et al., 2013*). The high quality of voltage clamp recordings in granule cells in vivo enabled detection of individual sensory-evoked EPSCs (*Chadderton et al., 2004*; *Rancz et al., 2007*; *Arenz et al., 2008*), thus providing exquisite sensitivity for detection of responses to a sensory stimulus. 45% of granule cells (60/133 cells) in crus I and II and 10% of those (3/30 cells) in the dorsal paraflocculus responded to somatosensory stimulation with a burst of EPSCs (*Figure 1A–E* left column and *Figure 1F–G*), as described previously (*Chadderton et al., 2004*; *Rancz et al., 2007*). In contrast, auditory stimulation (white noise, 81–91 dB SPL, *Figure 1A–E* middle column) evoked EPSCs in 25% of granule cells in crus I and II (33/133 cells) and 10% (3/30 cells) of granule cells in the dorsal paraflocculus (*Figure 1F–G*). When different sound levels were systematically tested, the number of evoked EPSC events increased with increasing sound levels (from 75 dB to 94 dB), while the mean amplitude of evoked EPSCs remained constant (*Figure 1—figure supplement 1*). Finally, visual stimulation (binocular LED flash; *Figure 1A–E* right column) evoked EPSCs in 87% of granule cells (26/30 cells) in the dorsal paraflocculus, while visual responses in crus I and II were rare (*Figure 1F–G*). When visual stimulation was delivered monocularly, ipsilateral stimulation evoked the predominant response (*Figure 1—figure supplement 2*). In summary, auditory and visual stimulation evoked bursts of EPSCs in granule cells that were comparable to those evoked by somatosensory stimulation (*Figure 1—figure supplement 3*).

## Multisensory responses in single cerebellar granule cells

A subpopulation of granule cells responded to multiple sensory modalities (*Figure 1F,G*). In dorsal paraflocculus 20% (6/30 cells) of granule cells were multisensory, and in crus I and II 14% (18/133 cells) were multisensory. In both regions, we found individual granule cells that responded to stimulation of three separate sensory modalities. *Figure 2* shows a representative granule cell in crus II that responded to auditory stimulation, somatosensory stimulation and a combination of these two stimuli.

To examine if responses to different sensory stimuli in a single granule cell were mediated by the same or separate mossy fibers, we analyzed the amplitude and waveform of individual EPSCs, since these characteristics reflect unique properties of distinct synapses (*Silver et al., 1996*; *Arenz et al., 2008*). When comparing EPSC amplitudes, we focused on the first EPSC of a sensory-evoked burst because the second and subsequent events are likely to be affected by synaptic facilitation and depression. In the cell shown in *Figure 2A–F*, the amplitude of auditory-evoked EPSCs (16.6 ± 0.8 pA, n = 23 sweeps) was significantly larger than that of somatosensory-evoked EPSCs (8.6 ± 1.2 pA, n = 23, $P < 0.05$), indicating that those two groups of EPSCs originate from different synapses, i.e. the signals are conveyed by different mossy fibers (Figure 4A). Across the population, 40% (8/20) of multisensory cells showed significantly different ($P < 0.05$) first EPSC amplitudes in response to different modalities (*Figure 2G–H* and *Figure 2—figure supplement 1*). Additionally, one cell also showed a significant difference ($P < 0.05$) in EPSC rise time for the different modalities. For the remaining cells that did not exhibit a significant difference in EPSC amplitudes between modalities, it is not possible to determine if their multimodal input arises from a single mossy fiber (Figure 4B) or from two mossy fibers that have indistinguishable characteristics (Figure 4A).

Combined stimulation of two modalities evoked responses that were approximately the sum of two unimodal responses (*Figure 2*). If the summation is perfectly linear, the linearity index (see *Figure 2* legend) should fall on the unity line. For the cell shown in *Figure 2*, this index was close to, but slightly below unity (*Figure 2C*). Interestingly, the linearity index showed considerable variation across cells and indicated moderately sublinear summation on average (0.71 ± 0.23; mean ± s.d.) for 8 cells that received inputs from two separate mossy fibers ( Figure 4C, see Discussion). The same tendency was seen for cells that were not determined to have separate mossy fiber inputs for different modalities (0.70 ± 0.20, mean ± s.d., n = 12), which was confirmed when the synaptic charge was used instead of event number to calculate the linearity index.

## Multisensory integration impacts action potential output

Finally, we examined how multisensory stimulation drives action potential output. We first identified granule cells receiving multisensory inputs via different mossy fibers using voltage-clamp recordings, and then we obtained recordings in the same neurons in current-clamp mode. The resting membrane potential and the action potential threshold were −55.0 ± 4.7 mV and −39.7 ± 3.0 mV (n = 4),

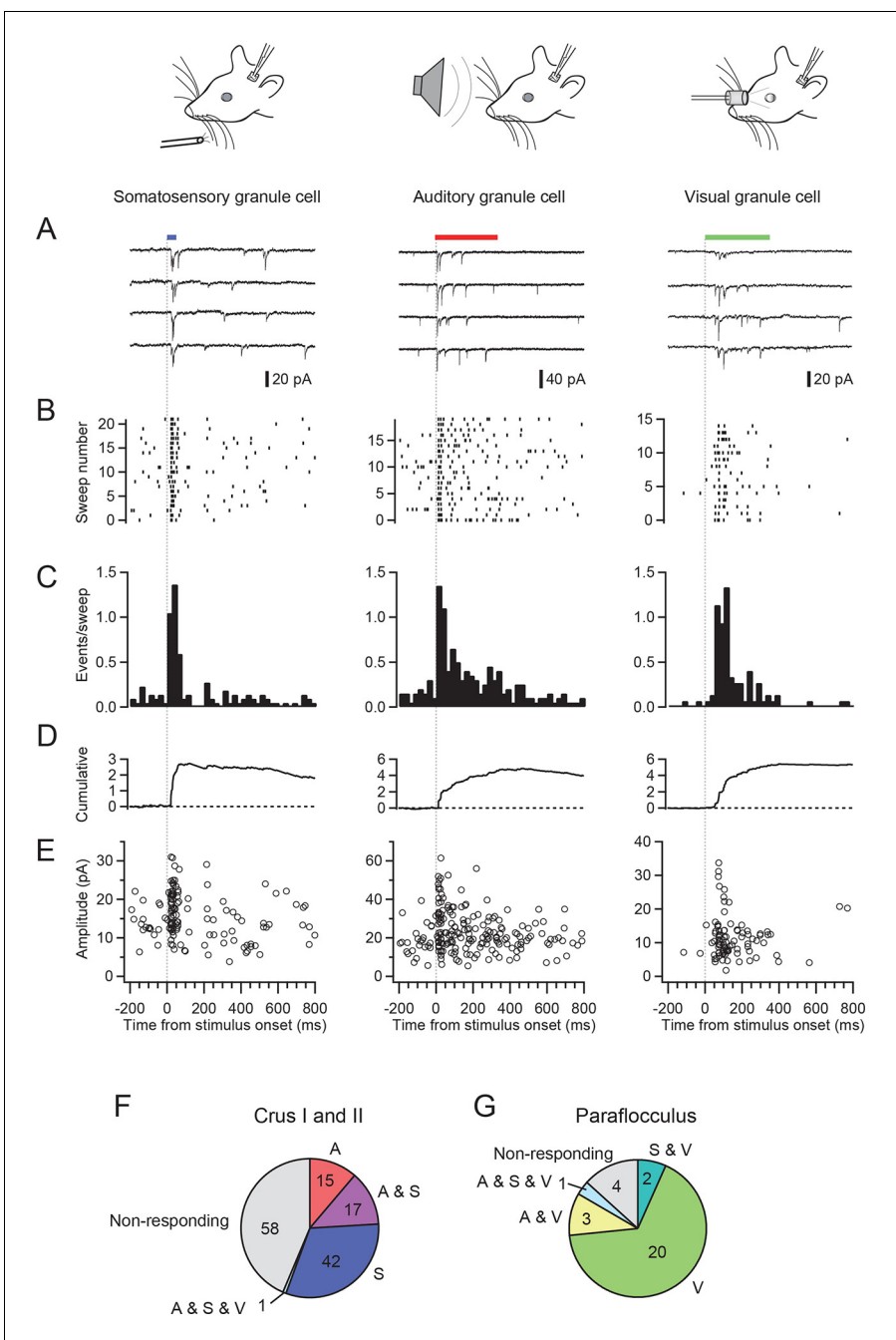

**Figure 1.** EPSCs evoked by activation of different sensory modalities in cerebellar granule cells in vivo. (A–F) Representative recordings from three different granule cells are presented for each modality. EPSCs were evoked by somatosensory stimulation (air puff to whiskers), auditory stimulation (white noise 89 dB) or visual stimulation (white LED, ipsilateral eye) recorded under voltage clamp at -70 mV. The bars above the traces indicate the timing of stimulation and the vertical dotted lines indicate the onset of the stimulation. The timescale is common for all panels A–E and is indicated at the bottom of panel E. (A) Four consecutive traces for each type of stimulation. (B) Raster plots of EPSC events. (C) Time histograms of EPSCs; bin width is 25 ms. (D) Cumulative time histograms (baseline subtracted). (E) The amplitudes of all EPSCs are plotted. (F–G) Pie charts indicating the total numbers of multimodal, unimodal and nonresponsive cells recorded in the crus I and II area (133 cells) and the dorsal paraflocculus (30 cells). Characters indicate the types of stimulation (S, somatosensory; A, auditory; V, visual), to which the cells responded. Cells that responded to multiple sensory modalities were indicated by the combination of those characters (A & S, S & V, A & V and A & S & V).

The following figure supplements are available for figure 1:

**Figure supplement 1.** Dependence of auditory responses of granule cells on sound levels (white noise).

*Figure 1 continued on next page*

*Figure 1 continued*

**Figure supplement 2.** The ipsilateral visual response was predominant over the contralateral response, as measured by the EPSC event number (A), the total synaptic charge (B) and the latency (C) evoked by visual stimulation by binocular LEDs (10 cells).

**Figure supplement 3.** Table summarizing the properties of EPSCs evoked by unisensory stimulation.

respectively. In the granule cell shown in *Figure 3A–C*, combined stimulation with two sensory modalities evoked more action potentials than the sum of two unimodal stimuli, indicating supralinear summation. Two of four cells exhibited such supralinear summation (Cell 1 and 2 in *Figure 3D*), while the other two cells showed sublinear summation. These findings suggest that, although there exists diversity across the population, granule cells are capable of integrating multisensory signals to generate enhanced action potential output.

## Discussion

We have taken advantage of the electrical compactness of cerebellar granule cells and their small number of synaptic inputs to probe how multisensory signals are integrated by single neurons at the input stage of the cerebellar cortex. Using high-resolution voltage-clamp recordings, we demonstrate directly that multisensory signals converge onto individual granule cells in vivo, and that multisensory input can enhance granule cell spike output.

### Multisensory integration in single granule cells

Granule cells receive excitatory input from only 4 mossy fibers on average (*Eccles et al., 1967*; *Jakab and Hamori, 1988*). Electrophysiological recordings have shown that somatosensory inputs to crus I and II (*Chadderton et al., 2004*; *Rancz et al., 2007*) and vestibular signals to the flocculus (*Arenz et al., 2008*) can be conveyed to individual granule cells by single mossy fibers. Therefore, it has been speculated that the other three mossy fibers (on average) could conduct signals of other sensory modalities. This conjecture has been supported by the recent anatomical (*Huang et al., 2013*) and in vitro electrophysiological (*Chabrol et al., 2015*) demonstration that single granule cells can receive mossy fibers of different origins (see also [*Sawtell, 2010*]). Our findings in vivo provide a direct functional demonstration that single granule cells can receive inputs from up to three separate sensory modalities. Furthermore, we show that combined stimulation of two sensory modalities can produce enhanced spike output from granule cells, indicating that the result of multisensory integration can be transmitted to downstream neurons in the cerebellar network.

While the present study strongly supports the conclusions of *Huang et al. (2013)* regarding multimodal integration in single cerebellar granule cells, we could not directly prove the integration of sensory and motor signals in granule cells as proposed by their study, because we could test only integration of sensory modalities in anesthetized animals. Integration of sensory and motor inputs should to be tested in future studies using recordings from granule cells in awake behaving animals (*Powell et al., 2015*).

When combining stimulation of two sensory modalities, we observed significant sublinear summation of synaptic currents in a subset of granule cells. Under the excellent voltage-clamp conditions that exist in granule cells, inputs from different synapses are expected to summate linearly. Thus, the observed sublinear summation of synaptic currents is likely to be due to inhibitory interactions between the two sensory pathways upstream from the granule cell (see *Figure 4*). Such interactions could occur at any point in upstream sensory pathways, including the brainstem, the thalamus and the cerebral cortex, particularly given that the relatively long latency (> 10 ms) of sensory responses in our recordings suggests that these sensory signals are mediated via the corticocerebellar pathway rather than direct projections from primary sensory neurons (*Morissette and Bower, 1996*). Further studies are required to understand the mechanism and significance of such inhibitory interactions.

It should also be noted that our findings do not directly contradict the absence of multimodal integration observed in granule cells in decerebrate animals in which the corticocerebellar pathway is not preserved (*Jörntell and Ekerot, 2006*; *Spanne and Jörntell, 2015*). It is likely that granule cells represent a diverse population with respect to functional multisensory input (*Figure 4*), with

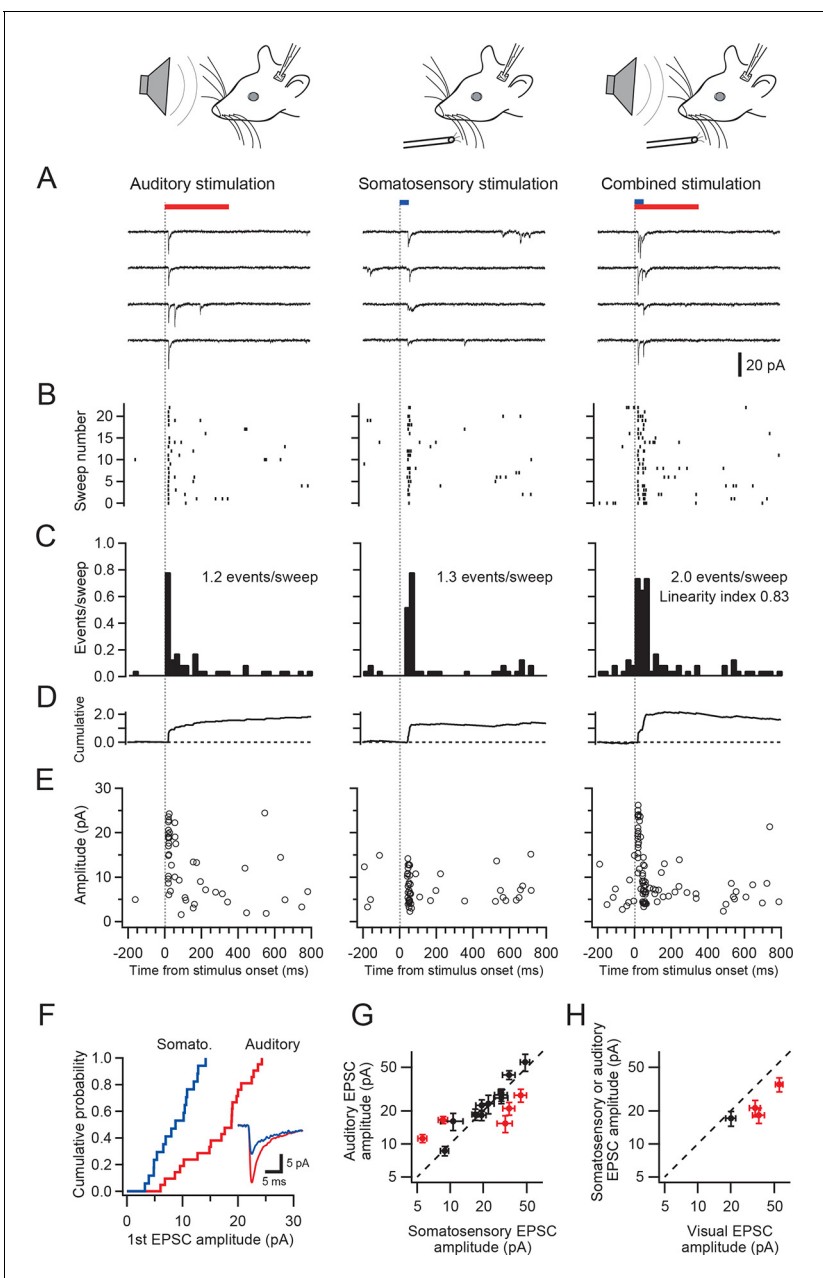

**Figure 2.** Single granule cells exhibit multi-sensory responses. EPSCs evoked by multi-sensory stimulation in a single cerebellar granule cell in crus II. EPSCs were evoked by somatosensory stimulation (air puff to whiskers), auditory stimulation (white noise, 91 dB) or a combination of the two. Trials were interleaved with an inter-trial interval of 3 s under voltage clamp at -70 mV. The color bars at the top indicate duration of stimuli, and the vertical dotted lines indicate the onset of the stimulation. The time scale is common for panels A–E and indicated at the bottom of E. (A) Four consecutive traces for each type of stimulation. (B) Raster plots of EPSC events. (C) Time histograms of EPSCs; bin width is 25 ms. The linearity index was calculated as the event number evoked by the combined stimulation divided by the sum of those evoked by the two unimodal stimulations. (D) Cumulative time histograms (baseline subtracted). (E) Amplitudes of all EPSCs. (F) Cumulative distribution of the amplitudes of the first events in a burst of evoked EPSCs. The amplitudes were significantly different between the two modalities ($P < 0.05$). Inset, average traces of first EPSCs. (G–H) Comparison of the amplitudes of EPSCs evoked by different sensory modalities. For each multimodal cell, the amplitudes of first-evoked EPSCs (mean ± s.e.m.) in response to two sensory modalities are plotted against each other. Eight out of 20 cells (16 in crus I and II [G] and 4 in dorsal paraflocculus [H]) had significantly different amplitudes ($P < 0.05$, indicated by red marks) and deviate from the unity line (dotted line).

The following figure supplement is available for figure 2:

**Figure supplement 1.** EPSC event numbers (A–B) and latencies (C–D) of the sensory responses of individual multimodal granule cells.

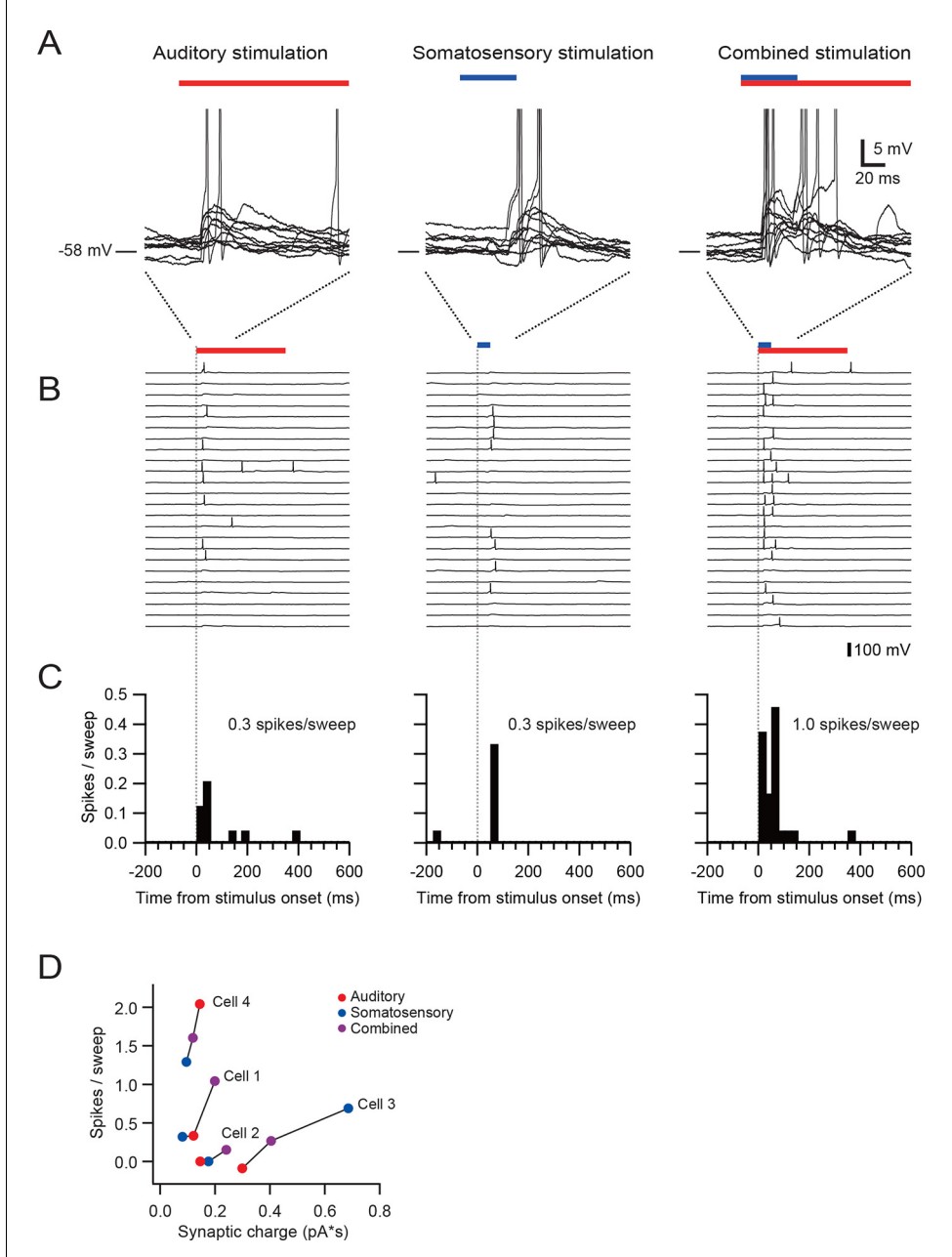

**Figure 3.** Multisensory integration can enhance granule cell output. (A–C) Action potentials in a representative granule cell evoked by multisensory stimulation. EPSPs and action potentials were evoked by somatosensory stimulation, auditory stimulation and combination of these two. Trials were interleaved with an inter-trial interval of 3 s. The granule cell was current-clamped with no bias current. The color bars at the top indicate the duration of stimulation and the vertical dotted lines indicate the onset of stimulation. (A) Representative traces are expanded to show evoked EPSPs and action potentials. Ten consecutive traces are overlaid. The peaks of action potentials are truncated. (B) All recorded traces are shown with the time scale indicated at the bottom of panel C. (C) Time histograms of evoked action potentials. The bin width is 25 ms. (D) Input-output relationships for 4 granule cells. The number of action potentials evoked in current-clamp mode was plotted against synaptic charge measured in voltage-clamp mode. The spike numbers are baseline-subtracted. Values from the same granule cell are connected by lines. Blue circles indicate the response to somatosensory stimulation, red circles auditory stimulation, purple circles combined somatosensory and auditory stimulation. The cell shown in A–C corresponds to Cell 1 in D.

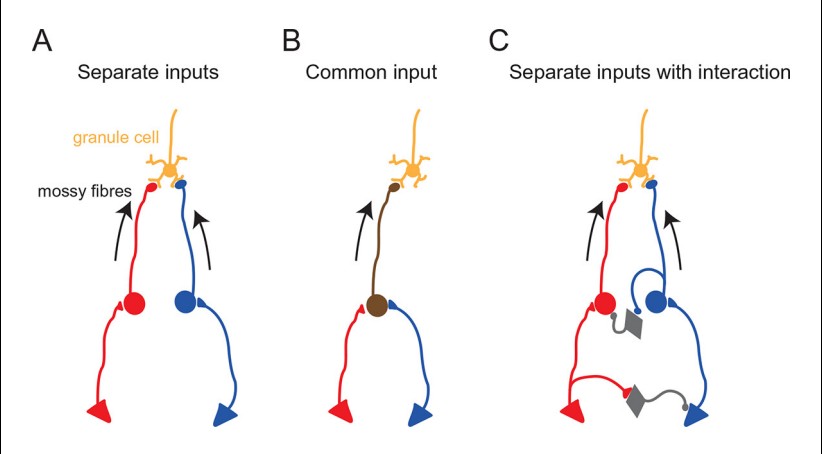

**Figure 4.** Functional configurations of multisensory integration at the mossy fiber–granule cell connection. Schematic diagrams showing potential anatomical substrates of the different multisensory integration scenarios described in the results. (**A**) Multimodal signals are transmitted by separate pathways and converge onto a single granule cell. (**B**) A single mossy fiber conveys mixed multi-modal signals. (**C**) Multimodal signals converge onto a granule cell, but the two pathways interact. In these schematics, the round cells represent pre-cerebellar neurons whose axons form mossy fibers. The triangular cells represent neurons projecting to the pre-cerebellar neurons (e.g. cortical neurons projecting to pontine neurons). Gray diamond-shaped neurons represent hypothetical interneurons. Another possibility for interaction between two separate pathways (not illustrated here) is presynaptic inhibition (*Mitchell and Silver, 2000*) or postsynaptic inhibition (*Duguid et al., 2015*) via Golgi cells.

some granule cells appearing to be unimodal, some with multisensory input delivered by separate mossy fibers, and some with multisensory input delivered by a single mossy fiber.

## Functional implications

The convergence of functionally distinct sensory signals in single granule cells at the input layer of the cerebellar cortex, predicted by *Huang et al. (2013)* and demonstrated directly here, is likely to be a crucial feature of cerebellar signal processing. Indeed, influential theories of cerebellar computation (*Marr, 1969*; *Albus, 1971*) have proposed that granule cells are not merely relaying signals but "recoding" multiple types of incoming signals. Our present study provides important in vivo functional evidence supporting this "recoding" hypothesis in the mammalian cerebellum. Moreover, it was hypothesized (*Albus, 1971*) that such "recoding" would expand the representation of population patterns because granule cells outnumber mossy fibers by a factor of 100 (due to the extensive branching of mossy fibers). Our observation that the linearity of summation (both in synaptic inputs and in spike outputs) varies across cells may reflect the diversity of coding patterns required for the computational role of the granule cell in expansion recoding. In future studies, it will be important to reveal how synaptic plasticity at the mossy fiber-granule cell synapse (*Roggeri et al., 2008*) may affect the representation of multiple sensory inputs at the level of a single granule cell.

## Materials and methods

All experiments were carried out in accordance with UK Home Office regulations and the guidelines of the Animal Experiment Committee of Jikei University. Lister-hooded rats (19–24 days old) were anaesthetized with a ketamine (60 mg/kg) and xylazine (4.5 mg/kg) mixture. Rats were freely breathing during surgery and recording. A peripheral anticholinergic drug, glycopyrrolate bromide (0.02 mg/kg S.C.) was administered in most of the experiments. In some early experiments, atropine (0.06 mg/kg) was used instead of glycopyrrolate. A head-post was glued onto the skull and a small craniotomy was made over the cerebellar region to be targeted. After removal of the dura, saline was used to prevent drying of the exposed brain surface.

Whole cell voltage-clamp (V-C) and current-clamp (I-C) recordings were made from granule cells in crus I and II and paraflocculus of the cerebellar cortex, using a Multiclamp 700B amplifier

(Molecular Devices, Sunnyvale, CA). The internal solution contained: K-methanesulphonate 133 mM, KCl 7 mM, HEPES 10 mM, Mg-ATP 2 mM, Na$_2$ATP 2 mM, Na$_2$GTP 0.5 mM, EGTA 0.05 mM and bio-cytin 0.5%, pH 7.2, giving an estimated chloride reversal potential (E$_{Cl}$) of -69 mV. This allows excitatory synaptic currents to be observed in isolation by voltage clamping at −70 mV. Data were low-pass filtered at 6 kHz and acquired at 50 kHz using a Digidata interface and pClamp software (Molecular Devices). Offline box smoothing (up to 11 points) was applied for noise reduction.

The animal was placed in a sound-attenuating light-proof box during the recording. Somatosensory stimulation was delivered using an air-puff (50 ms, 20–50 psi at source) timed by a Picospritzer and aimed at the ipsilateral whiskers, perioral skin or eye regions. Audible noise caused by the air-puff apparatus was carefully minimized and did not evoke cerebellar responses by itself. Auditory stimulation was delivered with a calibrated speaker driven by an RP2.1 processor and RPdvs software (Tucker-Davis Technologies, Alachua, FL). Gaussian white noise (up to 20 kHz) was presented for 350 ms with linear ramp rise and fall (5 ms). Visual stimulation was presented using two white LEDs (approximate intensity 200 mcd, one for each eye) each placed at 10 mm from the left or right eye. Each LED was light-shielded with a black cylinder, which also surrounds the eye, in order to deliver monocular stimulation. These LEDs diffusely illuminate a wide visual field because they are out of focus for the rat's vision. In a subset of visual experiments, a computer screen placed at 12 cm from the animal head was used to deliver a wide-field visual presentation (from 10° contralateral to 60° ipsilateral) of full screen flickering from black to white (10 frames/s).

The detection of EPSCs and action potentials was performed using a custom threshold-based algorithm programmed in Igor Pro (TaroTools: https://sites.google.com/site/tarotoolsregister/). The event number evoked by stimulation was counted (baseline-subtracted) in a time window adjusted for each cell to include all evoked events. Sensory responses were defined as positive when the post-stimulus histogram exceeded three times the standard deviation of the baseline. The synaptic charge was measured as the integral of the averaged current trace (baseline-subtracted and sign-reversed). Sensory response latency was defined as the time from the stimulus onset to the first EPSC event. In the analysis for *Figure 3D*, a granule cell that had an extremely large time difference (109 ms) between two unimodal responses compared to other cases (<35 ms) was excluded. Data are represented as mean ± s.e.m unless otherwise noted. Statistical significance was tested using the unpaired Student's t-test unless otherwise noted.

## Acknowledgements

We are grateful to Charlotte Arlt, Beverley Clark, Dimitar Kostadinov, Arnd Roth, Greg Stuart and Christian Wilms for helpful discussions and for comments on the manuscript. We thank Toshihiko Momiyama for his support, Hysell Oviedo and Jennifer Linden for help with setting up auditory equipment and Arifa Naeem for technical assistance. This work was supported by grants from the Wellcome Trust and the Gatsby Charitable Foundation (to MH) and Grant-in-Aid for Scientific Research from the Ministry of Education, Culture, Sports, Science and Technology of Japan and grants from the Uehara Memorial Foundation and the Takeda Science Foundation (to TI).

## Additional information

### Competing interests

MH: Reviewing editor, *eLife*. The other authors declare that no competing interests exist.

### Funding

| Funder | Grant reference number | Author |
| --- | --- | --- |
| Wellcome Trust | 094077 | Michael Häusser |

The funders had no role in study design, data collection and interpretation, or the decision to submit the work for publication.

## Author contributions

TI, MS, Conception and design, Acquisition of data, Analysis and interpretation of data, Drafting or revising the article; MH, Conception and design, Analysis and interpretation of data, Drafting or revising the article

## Ethics

Animal experimentation: This study was performed in strict accordance with UK Home Office regulations and under approval and supervision of the Animal Experiment Committee of Jikei University. Experiments were carried out under Project Licence 70/7833 issued by the Home Office, which was issued following local ethical review, and under the relevant Personal Licences issued by the Home Office. All surgery was performed under ketamine/xylazine anesthesia, and every effort was made to minimize suffering.

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
