## [Decision Letter]

Thank you for submitting your work entitled "Multimodal sensory integration in single cerebellar granule cells in vivo" for consideration by *eLife*. Your article has been reviewed by three peer reviewers, and the evaluation has been overseen by Eve Marder as the Senior Editor and Reviewing Editor.

The reviewers have discussed the reviews with one another and the Reviewing Editor has drafted this decision to help you prepare a revised submission.

As you know, we usually provide a summary of the needed revisions, but instead I am providing you with the entire set of reviewer comments. The revisions needed are exclusively editorial, and many of them relatively minor, but it seems that the context provided by these full reviews will help you understand what is necessary for you to directly address.

Reviewer #1:

It has long been thought that cerebellar granule cells perform multimodal integration and that such integration represents an important step in cerebellar processing, however whether or not granule cells actually perform such integration has been debated until recently. An elegant anatomical study published in e*Life* provided clear evidence that individual granule cells in some cerebellar regions integrate mossy fiber input from different sources. A recent study (Chabrol et al.) provides clear evidence for multimodal integration in cerebellar granule cells using both anatomy and in vitro electrophysiology. Multimodal integration has also been shown in vivo in granule cells in a cerebellum-like structure in electric fish. The present study by Ishikawa et al. provides further support for multimodal integration in cerebellar granule cells using in vivo electrophysiological recordings.

The central finding is that a subset of granule cells receive input about different sensory modalities and that in some cases these are conveyed by different mossy fibers (as indicated by distinct EPSC waveforms). This is a strong result with clear relevance to the field and provides in vivo physiological support for the results of Huang et al. (albeit in relation to different sets of inputs).

The authors also make claims regarding the important functional question of how multimodal convergence affects spiking output and about the different possible functional connectivity patterns in granule cells (unimodal, multimodal input conveyed by a single anatomical input etc). These claims are less well supported. The analysis of spiking output involves only four neurons amongst which half were suppressed and the other half enhanced by multimodal stimulation.

Discussion: …"Importantly, we show that granule cells represent a diverse population with respect to functional multisensory input, with some granule cells appearing to be unimodal, some with multisensory input delivered by separate mossy fibers, and some with multisensory input delivered by a single mossy fiber. These different anatomical and functional arrangements are illustrated schematically in Figure 3." I am not convinced that the authors have provided strong support for these claims. Some granule cells may be unimodal but the authors tested only a tiny fraction of the possible stimuli that could engage granule cells, so the appearance of unimodality does not count for much in my opinion. In the Results the authors say that for cases in which EPSCs don't differ significantly "it is not possible to determine if multimodal input arises from a single mossy fiber or from two mossy fibers that have indistinguishable characteristics." If this is true what is the support for the claim that some granule cells receive multisensory input delivered by a single mossy fiber? The statements made here need to be carefully qualified.

While the present manuscript is clearly relevant to the Huang et al. paper, the relationship between the two including the similarities/differences and advantages/limitations of both approaches could be made more explicit. This seems to warrant a separate paragraph in Discussion. For example Huang et al. looked at integration of cuneate and pontine (sensory and motor inputs) while the present paper looks at convergence of various sensory inputs.

*Reviewer #2:* In this manuscript, the authors accomplish the challenging task of making in vivo whole cell patch clamp recordings from cerebellar granule neurons in anesthetized rat. By doing so, they demonstrate conclusively that a subset of granule neurons can respond to two or more types of sensory stimulation (most frequently, auditory and somatosensory; 20% in the paraflocculus and14% in crus I & II). While this multisensory sensitivity is predicted from the anatomy, conflicting in vitro electrophysiological data make it important to demonstrate physiologically in an in vivo preparation. In fact, the result that the majority of cells were sensitive to just one modality (at least with the stimulus parameters tested) yet a significant minority were multisensory goes a long way toward resolving the discrepancy in the literature, although the authors (surprisingly?) don't emphasize this aspect of the work. The authors go on to show that in these multimodal cells, the EPSCs evoked by the two sensory stimuli applied together sum slightly sublinearly in voltage clamp, and evidence and reasoning are presented to suggest that this is indicative of upstream interactions in the two sensory pathways. The authors also demonstrate in current clamp that the sublinear EPSC summation can result in either more or fewer spikes than elicited by stimulation through a single modality, which is a useful additional piece of information, since it shows that multiple factors determine the nature of spike integration. The data are of high quality, and the manuscript is for the most part clearly written. It is hard not to wish for more manipulations, but given the technical difficulty of the recordings, it is not surprising that the work stops where it does. The manuscript seems well suited to *eLife*'s description of a Research Advance, since the paper by Hantman and colleagues that precedes this study was anatomical, and the present work offers a (much needed) physiological confirmation. All my comments are stylistic.

1) Abstract: "These findings provide functional evidence for convergence of multimodal signals onto single granule cells." That is true but somehow this seems to go just beyond "providing evidence" for multimodal signals. The multimodal signals are actually measured.

2) Introduction, "Therefore, determining the extent to which multimodal convergence is functionally relevant in vivo requires direct recordings from single granule cells using sensory stimulation of different modalities." This sentence is likewise a bit oblique and off the point, as there is nothing about functional relevance here. Isn't the question that whether granule cells in the intact, functioning brain respond to multimodal inputs can only be assessed with direct in vivo measurements in the species of interest?

3) Please indicate species at the beginning of Results.

4) Suggestion if space permits: please give a hint in the results of why the sublinear summation might be interesting. On first reading, it seemed that non-linear summation of synaptic inputs would be expected, unless the idea was that the different modalities on the different dendrites would necessarily be so segregated that they would have to sum linearly. The Discussion cleared this up, but a little more extensive orienting of the reader earlier would be helpful.

*Reviewer #3:* The manuscript describes powerful and straightforward measurements indicating that single cerebellar granule neurons receive multimodal sensory inputs; it also shows that such multimodal inputs are summed in a simple manner to generate spike output. These are striking results given that these neurons receive input from 3-5 mossy fiber presynaptic terminals. They are also important because they provide direct evidence for a hypothesized integrative role for granule cells in sensorimotor processing.

In general the paper is clearly written and reports interesting findings that should be of interest to a general audience of neuroscientists. I have no substantive concerns and recommend that the paper be accepted with minimal revisions.

Minor comment: The last sentence of the Introduction seems indirect. The authors might consider something like, "To determine whether multimodal convergence is common we made whole cell recordings in vivo from single granule cells and tested responsiveness to sensory stimulation of different modalities."

---

## [Author Response]

Reviewer #1:

*[…] Discussion:* …

*"Importantly, we show that granule cells represent a diverse population with respect to functional multisensory input, with some granule cells appearing to be unimodal, some with multisensory input delivered by separate mossy fibers, and some with multisensory input delivered by a single mossy fiber. These different anatomical and functional arrangements are illustrated schematically in Figure 3." I am not convinced that the authors have provided strong support for these claims. Some granule cells may be unimodal but the authors tested only a tiny fraction of the possible stimuli that could engage granule cells, so the appearance of unimodality does not count for much in my opinion. In the Results the authors say that for cases in which EPSCs don't differ significantly "it is not possible to determine if multimodal input arises from a single mossy fiber or from two mossy fibers that have indistinguishable characteristics." If this is true what is the support for the claim that some granule cells receive multisensory input delivered by a single mossy fiber? The statements made here need to be carefully qualified.*

We agree that these conclusions are rather speculative. We have removed this section, and added a more carefully worded statement later in the Discussion.

*While the present manuscript is clearly relevant to the Huang et al. paper, the relationship between the two including the similarities/differences and advantages/limitations of both approaches could be made more explicit. This seems to warrant a separate paragraph in Discussion. For example Huang et al.looked at integration of cuneate and pontine (sensory and motor inputs) while the present paper looks at convergence of various sensory inputs.*

A separate paragraph has been added to the discussion as suggested.

Reviewer #2:

*[…] The manuscript seems well suited to eLife's description of a Research Advance, since the paper by Hantman and colleagues that precedes this study was anatomical, and the present work offers a (much needed) physiological confirmation. All my comments are stylistic.*

We are grateful for the positive assessment of our work.

*1) Abstract: "These findings provide functional evidence for convergence of multimodal signals onto single granule cells." That is true but somehow this seems to go just beyond "providing evidence" for multimodal signals. The multimodal signals are actually measured.*

We have rephrased this to emphasize the actual measurement.

*2) Introduction, "Therefore, determining the extent to which multimodal convergence is functionally relevant in vivo requires direct recordings from single granule cells using sensory stimulation of different modalities." This sentence is likewise a bit oblique and off the point, as there is nothing about functional relevance here. Isn't the question that whether granule cells in the intact, functioning brain respond to multimodal inputs can only be assessed with direct in vivo measurements in the species of interest?*

This is a good suggestion – we have changed the last sentence of the introduction.

*3) Please indicate species at the beginning of Results.*

Added as suggested.

*4) Suggestion if space permits: please give a hint in the results of why the sublinear summation might be interesting. On first reading, it seemed that non-linear summation of synaptic inputs would be expected, unless the idea was that the different modalities on the different dendrites would necessarily be so segregated that they would have to sum linearly. The Discussion cleared this up, but a little more extensive orienting of the reader earlier would be helpful.*

This is an interesting suggestion, but such explanation would require significant description in the Results. Due to space limitations, we instead provide a pointer to the Discussion.

Reviewer #3:

The manuscript describes powerful and straightforward measurements indicating that single cerebellar granule neurons receive multimodal sensory inputs; it also shows that such multimodal inputs are summed in a simple manner to generate spike output. These are striking results given that these neurons receive input from 3-5 mossy fiber presynaptic terminals. They are also important because they provide direct evidence for a hypothesized integrative role for granule cells in sensorimotor processing. In general the paper is clearly written and reports interesting findings that should be of interest to a general audience of neuroscientists. I have no substantive concerns and recommend that the paper be accepted with minimal revisions.

We thank the reviewer for the positive feedback.

*The last sentence of the Introduction seems indirect. The authors might consider something like, "To determine whether multimodal convergence is common we made whole cell recordings in vivo from single granule cells and tested responsiveness to sensory stimulation of different modalities."*

This is a great suggestion – we have adopted this almost verbatim.